# Choline in Pediatric Nutrition: Assessing Formula, Fortifiers and Supplements Across Age Groups and Clinical Indications

**DOI:** 10.3390/nu17101632

**Published:** 2025-05-09

**Authors:** Wolfgang Bernhard, Anna Shunova, Ute Graepler-Mainka, Johannes Hilberath, Cornelia Wiechers, Christian F. Poets, Axel R. Franz

**Affiliations:** 1Department of Neonatology, University Children’s Hospital Tübingen, Faculty of Medicine, Eberhard-Karls-University, 72076 Tübingen, Germany; anna.shunova@med.uni-tuebingen.de (A.S.); cornelia.wiechers@med.uni-tuebingen.de (C.W.); christian-f.poets@med.uni-tuebingen.de (C.F.P.); axel.franz@med.uni-tuebingen.de (A.R.F.); 2Department of General Pediatrics, Hematology & Oncology, University Children’s Hospital Tübingen, Faculty of Medicine, Eberhard-Karls-University, 72076 Tübingen, Germany; ute.graepler-mainka@med.uni-tuebingen.de; 3Department of Pediatric Gastroenterology and Hepatology, University Children’s Hospital Tübingen, Faculty of Medicine, Eberhard-Karls-University, 72076 Tübingen, Germany; johannes.hilberath@med.uni-tuebingen.de; 4Department of Hematology and Oncology, University Children’s Hospital Tübingen, Faculty of Medicine, Eberhard-Karls-University, 72076 Tübingen, Germany; 5Center for Pediatric Clinical Studies, University Children’s Hospital Tübingen, Faculty of Medicine, Eberhard-Karls-University, 72076 Tübingen, Germany

**Keywords:** choline deficiency, cystic fibrosis, formula, intestinal disease, kidney disease, LC-PUFA, liver disease, pediatric nutrition, preterm infants, vitamins

## Abstract

**Background**: Sufficient choline supply is essential for tissue functions via phosphatidylcholine and sphingomyelin within membranes and secretions like bile, lipoproteins and surfactant, and in one-carbon metabolism via betaine. Choline requirements are linked to age and genetics, folate and cobalamin via betaine, and arachidonic (ARA) and docosahexaenoic (DHA) acid transport via the phosphatidylcholine moiety of lipoproteins. Groups at risk of choline deficiency include preterm infants, children with cystic fibrosis (CF) and patients dependent on parenteral nutrition. Fortifiers, formula and supplements may differently impact their choline supply. **Objective**: To evaluate added amounts of choline, folate, cobalamin, ARA and DHA in fortifiers, supplements and formula used in pediatric care from product files. **Methods**: Nutrient contents from commonly used products, categorized by age and patient groups, were obtained from public sources. Data are shown as medians and interquartile ranges. **Results**: 105 nutritional products including fortifiers, formula and products for special indications were analyzed. Choline concentrations were comparable in preterm and term infant formulas (≤6 months) (31.9 [27.6–33.3] vs. 33.3 [30.8–35.2] mg/100 kcal). Products for toddlers, and patients with CF, kidney or Crohn’s disease showed Choline levels from 0 to 39 mg/100 kcal. Several products contain milk components and lecithin-based emulsifiers potentially increasing choline content beyond indicated amounts. **Conclusions**: Choline addition is standardized in formula for term and preterm infants up to 6 months, but not in other products. Choline content may be higher in several products due to non-declared sources. The potential impact of insufficient choline supply in patients at risk for choline deficiency suggests the need for biochemical analysis of products.

## 1. Introduction

### 1.1. Essential Nature of Choline

Choline has been acknowledged as an essential nutrient by the Institute of Medicine (IoM) and the European Food Safety Authority (EFSA) since 1998 and 2016, respectively, because endogenous synthesis via the phosphatidylethanolamine-N-methyltransferase (PEMT) pathway is not sufficient to meet requirements [1,2]. Moreover, any endogenous choline synthesis by PEMT is partly fed by betaine which is derived from exogenous choline [3,4,5], and the release of free choline from PC synthesized by the PEMT pathway is low [6,7], irrespective of PEMT single nucleotide polymorphisms (SNPs, see below). In addition, choline has been approved by EFSA for the promotion of liver development and function, and clinical data highlight its role in human development and patient care [8,9,10]. Although EFSA provides health claim data on the choline metabolite betaine as well, betaine is not added to human formula and supplements [11]. We will therefore focus on choline administration in infants and children.

Choline is a critical nutrient in the overall population, particularly in several vulnerable groups at risk for deficiency, such as preterm infancy (PI), cystic fibrosis (CF) with exocrine pancreatic insufficiency, long-term total parenteral nutrition (TPN), small intestine bacterial overgrowth (SIBO) resulting in intestinal choline degradation prior to absorption, and frequent SNPs of the PEMT gene [7]. Such conditions and SNPs corrupting the estrogen-dependent expression of the PEMT pathway, result in a further increase in choline requirements and susceptibility to develop liver steatosis. In all these patients, plasma choline levels are frequently below their age-adjusted levels, impairing cellular choline uptake and potentially also organ function, particularly of the liver [12,13,14].

### 1.2. Choline Concentrations and Metabolism

The plasma levels of choline and its metabolite betaine indicate choline status, which depends on choline supply by regular nutrition, formula and nutritional supplements ([6], supplement) and [13,15,16]. Reference values of choline are age-dependent, being 41 (32–51) µmol/L in the fetus, 14 (10–17) µmol/L in pregnant women, and 9 (6–11) µmol/L in non-pregnant adults. Values display an untimely drop to 22 (16–28) µmol/L in preterm infants after birth during NICU stay and are only 6 (5–7) mmol/L in CF with exocrine pancreatic insufficiency and TPN patients. Values for betaine are 26 (18–39)µmol/L for the fetus and adult, but low in pregnant women (11 (10–14) µmol/L) and CF patients (19 (15–25) µmol/L) [6,17,18,19].

The PEMT pathway for endogenous choline synthesis requires betaine and further downstream metabolites of choline. It, therefore, depends on exogenous choline. Betaine is synthesized from choline in the kidneys for osmolytic functions, and in the liver to convert homocysteine to methionine, for the formation of S-adenosylmethionine (SAM). Notably, 40% of choline is metabolized to betaine, facilitating methyl group transfer rather than membrane phospholipid homeostasis. SAM is a substrate for the PEMT reaction, generating phosphatidyletanolamine-derived PC for very-low density lipoprotein (VLDL) formation, creatine synthesis, and epigenetic regulation of DNA and histones [9,20,21].

The functional impact of choline is that the concentrations of its major metabolites, phosphatidylcholine (PC) and sphingomyelin (SPH), are high in parenchyma as well as in bile, lipoproteins and other secretions, such as surfactant and gastric juice. PC concentrations in membranes and secretions are tightly regulated with characteristic fatty acid profiles [22,23,24]. PC and SPH synthesis essentially depend on exogenous choline supply [1,2,8] that, in addition to the consequences of PEMT SNPs, is further increased in CF patients due to choline losses and low PEMT expression, while PEMT expression is virtually absent in (preterm) infants [17,21,25].

### 1.3. Choline Function and Organotypic Metabolism

PC/choline concentrations have to be constant as any significant decrease, particularly of PC, results in organ damage [26,27]. Moreover, PC/choline is required for SPH synthase which converts ceramides to SPH via phosphocholine transfer, a process that is quantitatively important—particularly for lung metabolism and integrity [5,28,29]. An adequate choline supply is also required for acetylcholine synthesis, which is fundamental to synaptogenesis, the development and function of the brain and leucocytes [18,30,31]. Hence, choline exerts complex effects on organ integrity, immunology, and epigenetics. Notably, choline requirements are proportional to growth, with the highest defined postnatal adequate intake (AI) in term infants (TI) (17–18 mg/kg/d) [1]. Requirements may even be higher in fetuses and preterm infants (PI) due to their 3–4-times higher growth rate, as physiologic plasma levels are not achieved by current feeding strategies [17,19]. The same applies to CF and TPN patients, due to inadequate intake/administration or losses [6,12].

Turnover of lung surfactant PC is very low; only its PC secretion in contribution to systemic lipoprotein trafficking is high. However, the hepatic PC secretion rate is very high for both triglyceride emulsification and lipase activation via bile and for VLDL (20% PC) assembly to release triglycerides [28,32,33,34]. It surmounts 50% of the hepatic PC pool per day in adults and may be even higher at higher metabolic rates [35,36]. In essence, for choline, PC, SPH and betaine (see above) homeostasis, the organism requires adequate exogenous choline intake that is proportional to growth. It is further increased in case of choline losses, such as exocrine pancreatic insufficiency in CF [6]. Choline/betaine exhaustion rapidly results in liver disease like steatosis [37,38]. For survival during choline deprivation, systemic prioritization of the liver occurs at the expense of other organs such as the lungs [32,33,36].

### 1.4. Choline in Nutrition

Unprocessed food frequently contains high amounts of choline in the form of organic esters that are either water-soluble, such as phosphocholine and glycerophosphocholine (GPC) in milk, or lipid-soluble, such as PC, SPH and lyso-PC in unprocessed eggs, viscera, meat, fish and many cereals. Free choline is high in processed food and formula [39]. In formula, the physiological esters are not specifically added, but may originate from basic ingredients or emulsifiers rather than from specifically added components to meet regulatory requirements [40,41]. Notably, choline absorption and plasma kinetics as well as its intestinal degradation to trimethylamine (TMA) by the microbiota prior to absorption, increasing TMA oxide (TMAO) in plasma, differ among these compounds. PC, phosphocholine and glycerophosphocholine show no or low TMAO formation, whereas the frequently used choline bitartrate shows the highest rate of TMAO formation [42]. This may be particularly important for CF patients with exocrine pancreatic insufficiency frequently showing SIBO and high TMAO levels from infancy onwards [6].

For vulnerable groups, the addition of choline is under official regulation by European Union (EU) enactments, defining the choline content of formula to be 25–50 mg/100 kcal for term born infant formula from 0 to 6 months, whereas no supplementation is advised beyond 6 months of age [41]. However, guidelines and recommendations from IoM, EFSA and the European Society for Paediatric Gastroenterology, Hepatology and Nutrition (ESPGHAN), as well as scientific data, are not congruent here. The statement of the EU committee that beyond the age of 6 months, other foods and endogenous synthesis are sufficient, opposes both the recommendations of the IoM and clinical data, which indicate high requirements and low endogenous synthesis [41]. Furthermore, ESPGHAN statements on the minimal choline requirements of preterm infants are not in line with the IoM recommendations either [43]. Here, the minimum ESPGHAN recommendations for choline supply in preterm infants are only ~45% of the IoM recommendations for term infants, despite their 3–4 fold higher physiologic growth rate [1,44]. For other pediatric diseases, where large amounts of choline are lost via feces, as in cystic fibrosis (CF) with exocrine pancreatic insufficiency [6,37], official guidelines do not address choline as a critical nutrient [45]. This is despite the fact that in CF patients, low plasma choline and betaine levels are pathognomonic and associated with hepatosteatosis [9].

### 1.5. Aim of Study

As choline is a potentially critical nutrient, the aim of this study was to investigate the choline content of frequently used commercial pediatric nutritional products in Germany. Furthermore, as individual nutrient requirements change during development and in response to diseases, this was assessed in relation to total energy and nutrients closely linked to choline metabolism, such as folate, cobalamin, DHA, and ARA. We assessed their content in fortifiers and formula for preterm and term infants, in liquid food for toddlers, in supplements and in formulations designed for children with specific indications, such as cow milk allergy, food allergies, other gastrointestinal conditions including malabsorption and inflammation (e.g., Crohn’s disease), increased nutrient requirements or kidney disease. We focused on the product information concerning added choline together with macronutrients, folate, cobalamin (vitamin B12), ARA and DHA. We further evaluated whether additional, unquantified food components that contain choline may increase choline intake over the producers’ information.

## 2. Materials and Methods

### 2.1. Study Design

We collected information on fortifiers, formulas and other products that were either clinically used, prescribed, or available to clinical staff responsible for the respective patient groups from July 2024 to January 2025. This product selection encompassed insights from nurses, physicians and dietary staff members, and was based on their practical and potential applications according to clinical indications (neonatology, pediatric gastroenterology, CF) and common practices related to dietary acceptance (nurses, dieticians, CF, pediatric gastroenterology). All patient files from the CF outpatient clinic at the University Children’s Hospital Tübingen were reviewed for products used (Ethical approval by Ethics Committee, Medical Faculty, Tübingen University Clinic, No. 121/2020BO2 19 March 2020). All identified products and their components were compiled into an Excel sheet (Microsoft Office Professional Plus 2024, Microsoft Corporation, Redmond, WA, USA) for quantitative analysis.

### 2.2. Sample Collection

We collected the information on contents in energy, protein, carbohydrate, choline, folate, cobalamin (vitamin B12), ARA and DHA in the above-mentioned products used as formula, fortifiers, substitutes and supplements in our Neonatal Intensive Care Unit (NICU), CF outpatient clinic, department of gastroenterology, and nutritional support unit of the Children’s Hospital Tübingen.

Data were systematically grouped according to indications (Table 1) so that a total of 75 products, i.e., 3 fortifiers for preterm infants, 11 preterm infant formulas, 39 formulas for term infants from 0 to 6 months and 14 for infants from >6–12 months and 8 products for toddlers (12–36 months) were selected (Appendix A Table A1). For special indications, like galactosemia, lactose intolerance, increased energy requirements, tube feeding or kidney disease, an additional 30 products were identified (Appendix A Table A2).

### 2.3. Sample Characterization

Products were characterized according to their contents in macronutrients (energy; protein, carbohydrates, fat), micronutrients (choline, folate, cobalamin) and their content in long-chain polyunsaturated fatty acids (LC-PUFA), namely ARA and DHA acid (Table 2) according to the producers’ information, being available online. We additionally screened all selected products for components that essentially contain choline, either on the basis of skimmed milk or whey (mainly phosphocholine, GPC, or on the basis of emulsifiers containing phospholipids (PC, SPH and lyso-PC), such as soya lecithin and egg lipids. While this information was available via online links (see Appendix A Table A1 and Table A2), no information was provided on the amount of additional choline from such components.

### 2.4. Statistics

For descriptive statistics, data are expressed as median and interquartile range (IQR), as data were frequently not normally distributed. Group comparisons were performed of the indicated numbers of sample (N) with non-parametric testing, and correction for multiple testing according to Dunn, using GraphPad Instat^®^, version 3.0 (San Diego, CA, USA). Statistical significance was accepted at *p* < 0.05.

## 3. Results

An overview of evaluated products is shown in the online supplement (Table A1 and Table A2), showing their use for age and patient groups, and special indications as well as the amount of choline added, the component used (choline chloride or bitartrate), as well as the use of choline-containing emulsifiers. To none of the analyzed products, choline compounds prevalent in breast milk or other foods, like GPC, phosphocholine, PC or SPH [40] were added.

### 3.1. Energy

Energy density of products for term infants and toddlers was 65 to 68 kcal/100 mL of final product. It was marginally higher in products for preterm infants 73.5 (67.3–80.0) kcal/100 mL (*p* < 0.05) but partly was significantly higher in products for special indications like older patients suffering from CF, Crohn’s or kidney disease and others (Figure 1A; Appendix A Table A2). Notably, for preterm infant fortifiers, energy density was provided as the incremental energy per unit breast milk or formula, if fortified according to the manufacturers’ recommendations.

### 3.2. Macronutrients

Protein contents of formula (0–0.5 y) and add-on products for older term infants (0.5–1 y) were 8.1 (7.8–8.5)% and 7.8 (7.7–8.2)% of energy, respectively, but were higher for preterm compared to term infant formula (11.3 [11.8–13.8]%; *p* < 0.01). In products for older patients and for special diseases, values were more variable, according to indications. In preterm infant fortifiers, protein was 32.8 (24.5–38.3)% relative to energy (*p* < 0.01), as it was only added in 5–20% amounts to breastmilk or preterm infant formula. Except for preterm infant fortifiers, where fat surmounted carbohydrates by far (52.8 [45.1–55.3]% vs. 25.9 [14.6–27.9]%, respectively; *p* < 0.05), there was a balanced contribution of 41–46% fat and 42–49% carbohydrates in all products (Figure 1B).

### 3.3. Choline

Figure 2A–C displays the information on choline, folate and vitamin B12 (cobalamin) added to the product groups shown in Figure 1. Contents of choline added per 100 mL and per 100 kcal were almost identical in preterm and term infant (0–6 months) formula, with only marginal differences (Figure 2A). Added choline per 100 kcal was slightly lower in products for preterm compared to term infants (31.9 [27.6–33.3] vs. 33.3 [30.8–35.2] mg/100 kcal; *p* > 0.05), with minimum values of 20.5 vs. 23.0 mg/100 kcal. However, maximum value was highest (46.8 mg/100 kcal; O-HP-2, Humana) for PI formula.

Other products frequently did not contain added choline, with median zero values per 100 mg or per 100 kcal (PI fortifiers, PI and TI formula, products for toddlers, other supplements and products for special diseases [see above]) (Figure 1A). Maximum values were 17.3, 37.3, 22.1, 36.7 and 37.9 mg/100 kcal, respectively. In a total of 105, no choline was added to 27, choline chloride was added to 29 and choline bitartrate was added to 34 products, whereas no specification of added choline was provided in 15 cases.

However, in many products, soya lecithin or egg lipids were used as emulsifiers (Table 3) or whole and skimmed milk, as well as demineralized whey were added without exact quantification in the summary of product characteristics, prohibiting any precise estimation of total choline content, but providing information on the minimum choline supply by the respective products (Table 3 compared to Appendix A, Table A1 and Table A2): three out of 3 preterm infant fortifiers comprised milk lipids, lecithin and/or whole milk compounds. Five out of 11 preterm infant formulas contained such compounds, whereas among term infant formula (0–6 Months) 30 out of 39 products contained additional choline-containing ingredients, and in formula for 6–12 month olds it was 13 out of 14. For toddlers, it was 4 out of 8, and for special indications it was 16 out of 24 products.

There was a significant overlap of the products’ content or absence of both added choline and ingredients containing choline components, so that there are products containing precisely the indicated choline value from zero to 39 mg/100 kcal or more than indicated without any value given.

### 3.4. Choline Related Nutrients: Folate, Cobalamin and Long-Chain Polyunsaturated Fatty Acids

The low variability of folate (Figure 2B) and cobalamin (vitamin B12; Figure 2C) values was in sharp contrast to added choline, showing no median zero values and much narrower IQRs for both. Only exceptions were high maximum values in preterm infant fortifiers, and supplements for older children. No significant differences were observed between groups (*p* > 0.05). Notably, although bioavailability differs for different components, such as cyano-, hydroxy- or methylcobalamin, no specific information was provided here. For folate, it was recorded that Bio Combiotic PRE, HA Combiotic PRE-HA, Bio Combiotic 1+2 and HiPP Comfort (all HiPP GmbH & Co, Pfaffenhofen, Germany) contained 5-methyltetrahydrofolate rather than folate.

For the (conditionally) essential LC-PUFA, ARA and DHA, values were similar in all formulas for preterm and term infants up to 6 months of age. However, for ARA from 6 months onwards, and for DHA from 1 year onwards, the values showed extreme ranges, including even zero-values for some products (e.g., high energy) (Figure 3). In toddler products and special health issues, ARA and DHA values were frequently zero, but could also be high (30.8 mg/100 kcal ARA and DHA, Beba Expert HA; 29.9 mg/100 kcal ARA and DHA, Beba AR-anti reflux; both Nestlé). Consequently, although being metabolically intimately connected with choline, the ratio of added choline vs. folate, cobalamin, ARA and DHA was highly variable throughout, with the lowest variability in formulas for preterm and term infants up to 6 months. However, ratios between added choline and folate, cobalamin, ARA and DHA only partly reflected the real situation as information on additional choline-containing compounds was not provided (Appendix A Figure A1, Appendix B).

## 4. Discussion

In this study, we investigated preterm and term infant fortifiers and formulas, as well as products for older children and for children with additional nutrient requirements due to an underlying disease. Our main focus was on the supply of choline, as an essential nutrient per se, and as a critical nutrient under special conditions. Such conditions principally are an increased requirement to achieve and maintain concentrations of choline and its primary water-soluble metabolite betaine in plasma that match age-adjusted reference levels. We evaluated this in relation to other macronutrients, such as energy and fractions of protein, carbohydrate and fat. Moreover, we investigated the other main micronutrients that are related to choline metabolism, namely folate and cobalamin which are involved in one-carbon metabolism via betaine as a choline metabolite, as well as ARA and DHA that are linked to choline via the PC moiety of very low-density lipoproteins (VLDL) as the main plasma carrier of these fatty acids (for details see Figure 4 and [19,40]).

### 4.1. Balance and Imbalance of Choline Compared to Other Nutrients

All these latter components were present in balanced concentrations and fractions, with only few exceptions: there were lower concentrations of energy in final dilutions of preterm infant fortifiers, being an add-on to formula or breast milk, and a higher fraction of protein and fat in such products. Moreover, ARA and DHA were partly absent from formula beyond 6 months of age, and in ‘other products’ compensating increased energy expenditure or those for characteristic intestinal (such as food allergies, malabsorption or Crohn’s disease) or kidney diseases. For folate and cobalamin, whose metabolism is tightly linked to that of choline due to their function as methyl group carriers for betaine and other downstream products, concentrations were similar and balanced throughout (see Section 3, Figure 2B,C).

However, this is in contrast to choline, both for the amounts of added choline per 100 mL volume and per 100 kcal supplied with a product. Some products for infants older than 6 months comprised no added choline, whereas others did. While infants beyond 6 months are frequently fed with complementary solid food already, their choline supply from formula may be insufficient, as their requirement per kg body weight is nearly identical to that of younger infants [1].

The only constancy in choline added to products, as officially regulated [41], is found in formula for PI and TI up to 6 months of age, comprising 30–35 mg added choline per 100 kcal. However, the results of our own research indicate that the choline amounts added to TI formula are not sufficient for PI. Under current feeding conditions, with both breast milk and formula, PI do not show plasma choline and betaine corresponding to fetal levels (postnatal PI plasma: ~20 µmol/L vs. cord blood: ~40 µmol/L). As the uptake of choline into cells is proportional to concentration in a range of ~2–100 µmol/L [27], PI formula may be inadequate for optimal growth. Notably, this applies similarly to unfortified breast milk that supplies the same median amount of choline per 100 kcal as PI formula, however with a much larger intra- and inter-individual variability [46].

A similar objection may be raised for products used for special clinical conditions such as CF and other diseases. We recently demonstrated that in CF patients with exocrine pancreatic insufficiency, plasma choline levels are significantly decreased and they further decrease with age and such low plasma choline values are associated with liver disease [6,9,38]. Products frequently used in these patients to compensate for increased energy expenditure have a substantial impact on plasma choline concentrations, if they contain additional choline ([6], Appendix A, Table A1 and Table A2). In essence, the choline content of such products is important to normalize plasma concentrations, and using an adequate product is a matter of knowledge and choice. From a clinical point of view, this is very important as there are strong correlations between plasma choline and its supply, and liver and lung function [9,21]. This may similarly apply for products used in other patient groups, e.g., d those with pediatric Crohn’s (e.g., Modulen IBD^®^, Nestlé), cachexia (Duocal^®^, Nutricia), liver (Heparon Junior^®^, Nutricia) or kidney disease (Nephea Infant^®^, Nephea Kid^®^, MetaX; restoric^®^ nephro prae, Vitasyn medical) (see Appendix A, Table A2).

### 4.2. The Indicated Choline Content May Not Match the Actual Amount in the Products

However, there is a discrepancy between the amounts of ‘intentionally added choline’ and that ‘actually present’ in some products (i.e., that choline form lecithin emulsifiers or milk components), which applies to all product groups from preterm infant fortifiers to special needs products. If added, mostly choline bitartrate, followed by choline chloride, is a quantified product ingredient. It is noteworthy that choline bitartrate, followed by choline chloride, is the least favourable compound as its conversion to TMA and TMAO surmounts that of other compounds, such as GPC, phosphocholine and PC [20,42]. However, 71 out of 105 products comprised ingredients that essentially contain choline esters, and potentially free choline mainly due to processing. These are (1) milk and whey, primarily containing GPC and phosphocholine, and not found in non-milk-based products for allergy reasons, and (2) phospholipid-based emulsifiers, like soya lecithin and egg lipids. Whereas egg lipids mainly contain PC as a phospholipid, soya lecithin is a mixture mainly of PC, phosphatidylethanolamine and phosphatidylinositol [47]. All these phospholipids do not increase TMAO formation [20,42].

### 4.3. Clinical Suggestions for Product Choice

While the amounts of choline added by milk components and lecithin-based emulsifiers are not indicated in the respective products (Appendix A, Table A1 and Table A2), they may substantially improve the choline supply to patients. Hence, for all patients at risk of choline undersupply or deficiency, that is preterm infants, CF patients, and patients of any age with malabsorption or dependence on parenteral nutrition, the choice of products comprising additional choline from milk or PC-containing emulsifiers may be useful. For example, products that do not contain choline chloride or bitartrate, may nevertheless be useful for CF patients with exocrine pancreatic insufficiency, as they may contain choline via lecithin (Table 3).

This holds true even though their quantitative contribution to choline supply is yet unknown, because of the very high upper tolerable limit of intake according to IoM and EFSA, and the virtual absence of choline toxicity (if not given intravenously as a high-dosage bolus injection) [1,2,8].

While choline supply is frequently higher than estimated from the provided information, when using such products, it may vary. In a previous analysis, two batches of a preterm infant formula, PC was the major choline component, and choline levels varied by 50%, reaching the upper limit of the ESPGHAN recommendation in one batch [40,43,44]. To the best of our knowledge, no choline toxicity has been reported for any of these products providing additional choline. Nevertheless, a declaration of choline from compounds like emulsifiers, and a straightforward biochemical analysis of formulas, fortifiers and add-on formulations would be helpful for clinicians and dieticians to enable informed choices. Compared to the value of a single batch, such costs seem negligible. Furthermore, replacing phospholipid emulsifiers with choline-free fatty acid methyl esters and similar compounds by companies is discouraged by these authors.

### 4.4. Limitations

While we used the most recent information on products, the market is fluctuating, and product characteristics change rapidly. Moreover, we have not evaluated the global market but focused on products being prescribed to or used by patients visiting our local outpatient clinic or during clinical treatment. However, the declared choline content, and its relation to folate, cobalamin, and long-chain polyunsaturated acids may be incorrect for those fortifiers, formula, liquid food and special nutrition products containing milk or lecithin compounds, adding natural choline components which are not quantified in the product information.

## 5. Conclusions

The amount of the essential nutrient choline in commercial pediatric nutritional products is heterogenous, particularly for products for children > 6 months of age. This study provides a comprehensive analysis of 105 nutritional products used in children regarding their choline content. However, choline may frequently be higher in products containing milk constituents and/or lecithin. Their contribution to choline supply, however, is unclear. Such uncertainty may have implications on product choice for children at risk of choline deficiency, such as preterm infants, and patients suffering from CF, SIBO, malabsorption or Crohn’s disease, receiving exclusive diets or are TPN-dependent.

In perspective, for all products used in patients at risk for choline deficiency, we propose a mandatory characterization or biochemical analysis to assess their total choline content, based on all contributing components.

## Figures and Tables

**Figure 1 nutrients-17-01632-f001:**
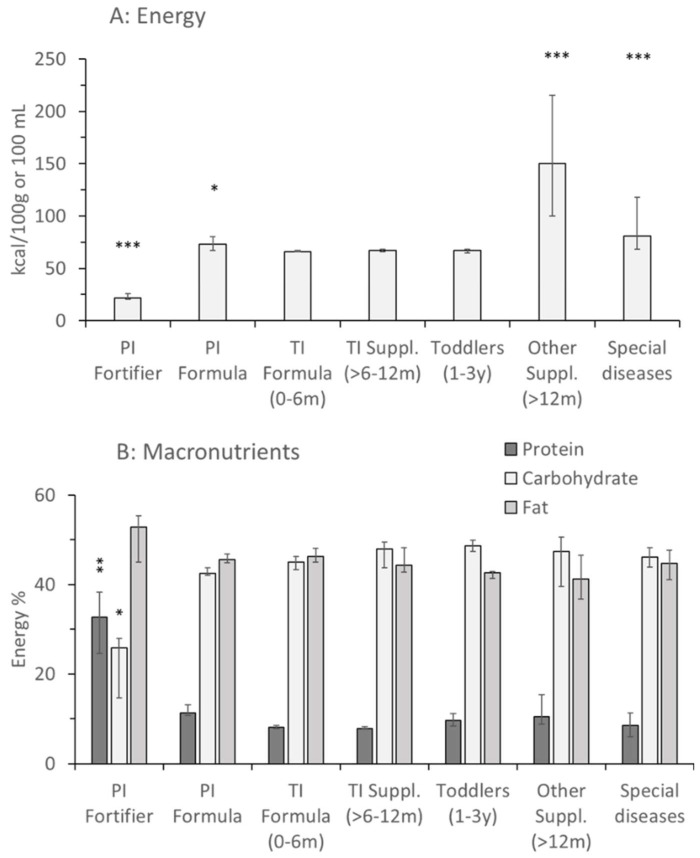
Energy concentration and macronutrient distribution of fortifiers, formulas and supplements for healthy children and pediatric patients. Data are median values and interquartile ranges of preterm infant (PI) fortifiers (*n* = 3), PI formula (*n* = 14), term infant (TI) formula (0–6 months [m]; *n* = 38), TI supplements (6–12 m; *n* = 14), toddlers (1–3 years [y]; *n* = 9), other supplements for higher energy demands, e.g., in cystic fibrosis (M = 19) and special indications like kidney disease, Crohn’s disease, etc., (*n* = 12). Values are median and interquartile range. Abbreviations: *, *p* < 0.05; **, *p* < 0.01; ***, *p* < 0.001 vs. TI formula 0–6 m.

**Figure 2 nutrients-17-01632-f002:**
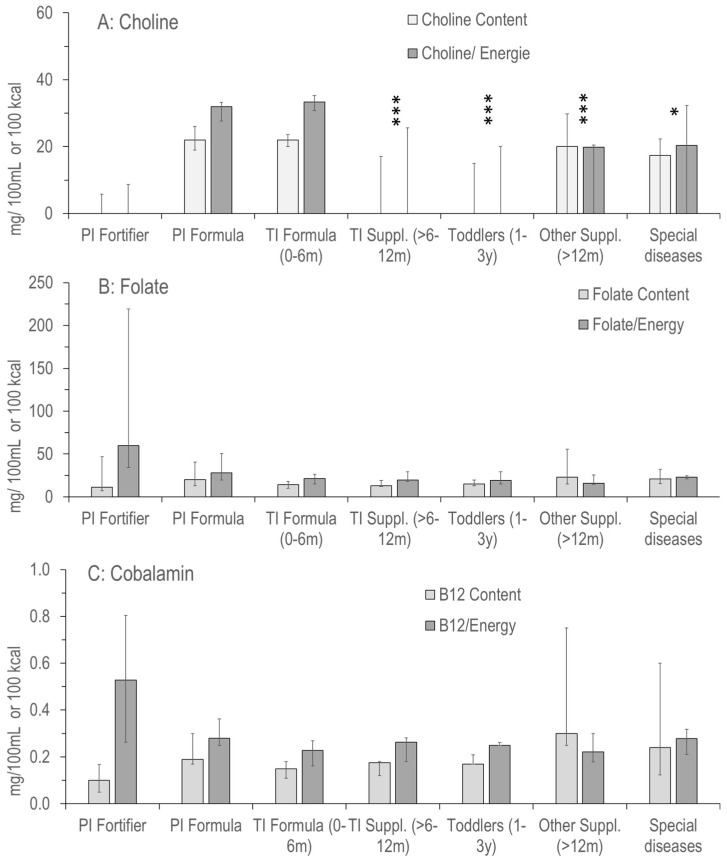
Concentration and density of choline (**A**) and its metabolically related vitamins folate (**B**) and vitamin B12 (**C**) in the products as indicated in Figure 1. Abbreviations: PI, preterm infant; TI, term infant; ***, *p* < 0.001; *, *p* < 0.05 vs. PI and TI Formula.

**Figure 3 nutrients-17-01632-f003:**
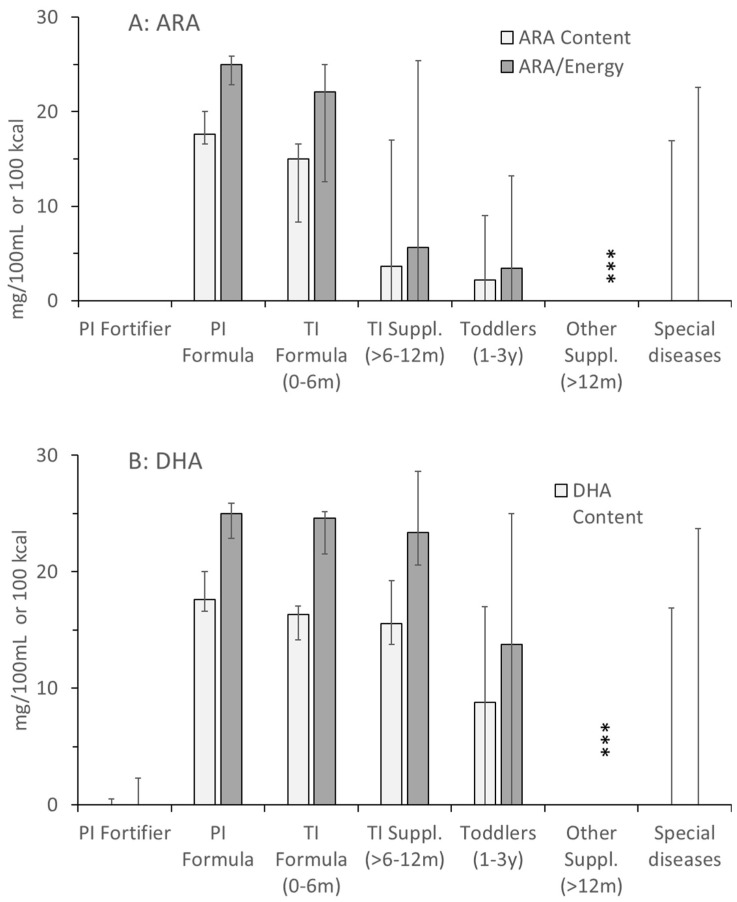
Concentrations and densities of arachidonic (ARA; (**A**)) and docosahexaenoic acid (DHA, (**B**)) in the products as indicated in Figure 1. Abbreviations: PI, preterm infant; TI, term infant; ***, *p* < 0.001.

**Figure 4 nutrients-17-01632-f004:**
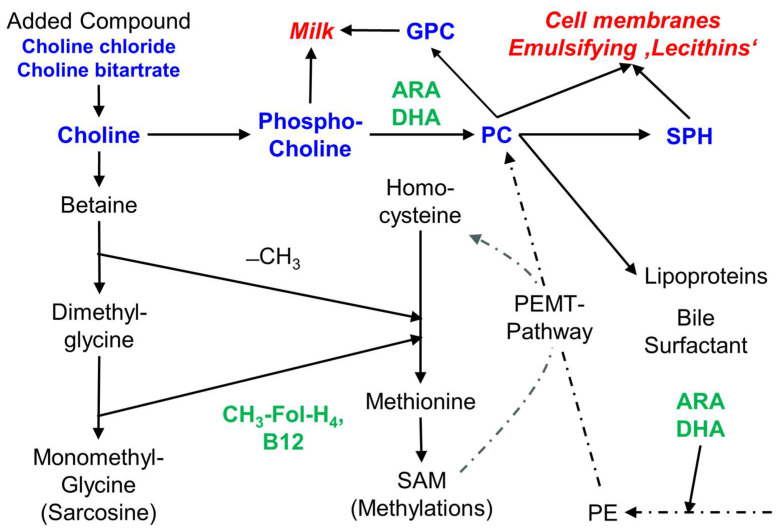
Scheme of choline components and metabolism, and the relation of choline to folate, vitamin B12, ARA and DHA. Choline and choline-containing components are bold in blue, choline-related micronutrients are bold in green. Abbreviations: ARA, arachidonic acid; B12, vitamin B12 (cobalamin); −CH_3_, methyl group; CH_3_-FolH_4_, methyl tetrahydrofolate; DHA, docosahexaenoic acid; GPC, glycerophosphocholine; PC, phosphatidylcholine; PE, phosphatidylethanolamine; PEMT, PE-N-methyl transferase; SPH, sphingomyelin.

**Table 1 nutrients-17-01632-t001:** Product groups assessed in study.

Patient Group	Preterm Infant Fortifier Preterm Infant Formula Term Infant Formula (0–6 months) Term Infant Supplementary Food (>6–12 months) Toddler Supplementary Food (12 months to 3 years;) Supplements in Cystic Fibrosis (>12 m) Products with other Indications

**Table 2 nutrients-17-01632-t002:** Compounds assessed in study.

Macronutrients	Energy (kcal/100 mL)	
	Protein (Energy-%)	
	Fat (Energy-%)	
	Carbohydrates (Energy-%)	
Micronutrients	(mg/100 mL)	(mg/100 kcal)
	Choline	Choline
	Folate	Folate
	B12	B12
	ARA	ARA
	DHA	DHA
Micronutrient Ratios	Choline/Folate; Choline/B12; Choline/ARA; Choline/DHA	

Abbreviations: ARA, arachidonic acid; B12, vitamin B12 (cobalamin); DHA, docosahexaenoic acid.

**Table 3 nutrients-17-01632-t003:** Products with choline-containing emulsifiers and milk components.

Product	Choline Added (mg/100 kcal)	Choline Component	Choline Containing Emulsifiers	Water Soluble Choline Components from Milk
**Preterm Infant Fortifier**				
BEBA FM 85, Nestlé *	17.3	No specification	Soya Lecithin	∅
Humavant+4, Prolacta	0	∅	Milk Lipids	Human Milk
Aptamil FMS, Nutricia	0	∅	Egg and Milk Lipids	∅
**Preterm Infant Formula**				
Aptamil HA Pre, Nutricia	33.3	Choline chloride	egg lipids	∅
Aptamil PDF, Nutricia	31.9	Choline chloride	egg lipids	Skimmed milk
Aptamil Prematil, Nutricia	32.5	Choline chloride	egg lipids	Skimmed milk
Aptamil Prematil HA, Nutricia	32.5	Choline chloride	egg lipids, soya lecithin	∅
Monogen, Nutricia	23.0	Choline chloride	∅	Skimmed milk
**Term Infant Formula (0–6 months)**				
Aptamil HA Pre, Nutricia	33.3	Choline chloride	egg lipids	∅
Aptamil PDF, Nutricia	31.9	Choline chloride	egg lipids	∅
Aptamil Prematil, Nutricia	33.3	Choline chloride	egg lipids	∅
Aptamil Prematil HA, Nutricia	31.9	Choline chloride	egg lipids, soya lecithin	∅
Bebita 1, Bebivita	38.8	No specification	∅	Skimmed milk
Bio Combiotic PRE, Hipp	37.9	No specification	∅	Skimmed milk
Bio Combiotic 1, Hipp	38.8	No specification	∅	Skimmed milk
Humana Anfangsmilch pre, Humana	30.8	Choline bitartrate	∅	Skimmed Milk, whey powder
Humana Anfangsmilch 1, Humana	37.3	Choline bitartrate	∅	Skimmed Milk, whey powder
kendamil first instant milk, Kendamil	32.8	Choline bitartrate	Whole milk lipids	skimmed milk, whey powder
Kendamil first organic, Kendamil	30.3	Choline bitartrate	Whole milk lipids	skimmed milk, whey powder
Kendamil first goat, Kendamil	32.8	Choline bitartrate	Whole goat milk lipids	skimmed goat milk, whey powder
Milumil Pre Anfangsmilch, Milupa	36.4	Choline chloride	Soya lecithin	Skimmed milk, whey components
Milumil 1 Anfangsmilch, Milupa	32.4	Choline chloride	Soya lecithin	Skimmed milk
Alfaré, Nestlé	29.9	Choline bitartrate	Soya lecithin	∅
BEBA Optipro PRE, Nestlé	30.8	Choline bitartrate	Soya lecithin	Skimmed milk, whey
BEBA Optipro 1, Nestlé	30.7	Choline bitartrate	Soya lecithin	Skimmed milk, whey
Aptamil Care Pre, Nutricia	34.8	Choline chloride	Soya lecithin	Skimmed milk
Aptamil Comfort, Nutricia	33.3	Choline chloride	Soya lecithin	∅
Aptamil Pronutra Pre, Nutricia	33.3	Choline chloride	Soya lecithin	Skimmed milk
Aptamil FMS, Nutricia	0	∅	Egg lipids	∅
Aptamil Anti Reflux, Nutricia	33.3	Choline chloride	Soya lecithin	Skimmed milk, whey
Aptamil Care Pre, Nutricia	34.8	Choline chloride	Soya lecithin	Skimmed milk, whey
Aptamil Bio PRE, Nutricia	33.3	Choline chloride	Lecithin, not specified	Skimmed milk, whey
Aptamil Organic PRE, Nutricia	33.3	Choline chloride	∅	Skimmed milk, whey
Aptamil Organic 1, Nutricia	33.3	Choline chloride	Soya lecithin	Skimmed milk, whey
Holle Bio Pre, Holle	33.3	Choline bitartrate	∅	Skimmed milk, whey
Bio-Anfangsmilch, Holle	33.8	Choline bitartrate	∅	Skimmed milk, whey
Bio-Anfangsmilch 1, Holle	33.3	Choline bitartrate	∅	Skimmed milk, whey
Bio-Anfangsmilch 1 aus Ziegenmilch, Holle	32.4	Choline bitartrate	Goat milk lipids	Whey
**Term Infant Supplements (6–12 months)**				
Bebivita 2 Folgemilch, Bebivita	0	∅	Lecithins, not specified	Skimmed milk
Bio Combiotic 2, Hipp	0	∅	∅	Skimmed milk, whey
HiPP Bio Ziegenmilch 2, Hipp	0	∅	∅	Skimmed goat milk
Humana Folgemilch 2, Humana	37.3	Choline bitartrate	∅	Skimmed milk, whey
Humana Folgemilch 3, Humana	37.3	Choline bitartrate	∅	Skimmed milk, whey
Follow-on milk, Kendamil	32.8	Choline bitartrate	Whole Milk lipids	Skimmed milk, whey
kendamil goat, Kendamil	24.2	Choline bitartrate	Whole Goat Milk Lipids	Skimmed Goat Milk, Whey
Milumil 2 Folgemilch, Milupa	25.0	Choline chloride	Soya lecithin	Skimmed milk, whey
Milumil 3 Folgemilch, Milupa	25.0	Choline chloride	Soya lecithin	Skimmed milk, whey
BEBA Optipro 2, Nestlé	0	∅	Soya lecithin	Skimmed milk, whey
Aptamil Bio 2, Nutricia	0	∅	Lecithins, not specified	Skimmed milk, whey
Monogen, Nutricia	23.0	Choline chloride	∅	Skimmed milk
BEBA Optipro 2, Nestlé	0	∅	Soya lecithin	∅
**Toddler Milk (12–36 Months)**				
Humana Kindergetränk 1+, Humana	0	∅	∅	Skimmed milk
Toddler milk, Kendamil	0	∅	Whole milk lipids	Skimmed milk
Toddler milk Organic, Kendamil	0	∅	Whole milk lipids	Water soluble Milk Components, Whey
BEBA Junior 12–36, Nestlé	0	∅	Milk fat, soya lecithin	Skimmed milk, whey
**Special Indications**				
Aptamil Profutura DUOADVANCE 2, Nutricia	22.1	Choline chloride	Milk fat, soya lecithin	Skimmed milk, whey
Energea Pkid, MetaX	0	∅	Sunflower lecithin	Skimmed milk
Energea P, MetaX	0	∅	Sunflower lecithin	Skimmed milk
Nephea Kid (Kidney Disease), MetaX	0	∅	Soya lecithin	∅
Nephea Infant (Kidney Disease), MetaX	0	∅	Soya lecithin	∅
Peptamen (Tube feeding), Nestlé	21	Choline chloride	Soya lecithin	∅
Peptamen Junior Advance (high energy tube feed), Nestlé	16	Choline chloride	Soya lecithin	∅
Aptamil PDF, Nutricia	31.9	Choline chloride	Egg lipids	Skimmed milk
Fortimel Joghurt Style	36.7	Choline chloride		Skimmed milk
Fortimel Pulver Neutral,	33.1	Choline chloride	Soya lecithin	∅
Infatrini 125 mL, Nutricia	31.3	Choline chloride	Soya lecithin	Skimmed milk
Nutrini Multi Fibre, Nutricia	20	Choline chloride	Soya lecithin	∅
NutriniDrink Compact Multi Fibre, Nutricia	20	Choline chloride	Soya lecithin	∅
Nutrini creamy, Nutricia	20	Choline chloride	Soya lecithin, milk lipids	Skimmed milk
restoric^®^ nephro prae, Vitasyn medical	0	∅	Soya lecithin	∅
restoric^®^ supportiv S (Malnutrition), Vitasyn medical	0	∅	Soya lecithin	∅

*, companies’ locations: Bebivita: Reisgang, Germany; Hipp: Pfaffenhofen, Germany; Holle Europe: Lörrach, Germany; Humana: Bremen, Germany; Kendamil: Kendal, UK; Meta X: Friedberg, Germany; Milupa: Frankfurt, Germany; Nestlé: Vevey, Switzerland; Nutricia: Frankfurt, Germany; Prolacta: Rochester, MN, USA; Vitasyn medical: Berlin, Germany.

## Data Availability

The original contributions presented in this study are included in the article and Appendix A and Appendix B. Further inquiries can be directed to the corresponding author.

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
