# Peer review of "Choline in Pediatric Nutrition: Assessing Formula, Fortifiers and Supplements Across Age Groups and Clinical Indications"

_nutrients, 2025, doi:10.3390/nu17101632_

Round 1

Reviewer 1 Report

Comments and Suggestions for Authors

In the current study the authors investigated the choline content and related nutrients (folate, cobalamin, arachidonic acid and docosahexaenoic acids of frequently used commercial pediatric nutritional products in Germany, namely 105 products. They found that the amount of choline in commercial pediatric nutritional products is heterogenous, particularly for products for children >6 months of age. This may have implications on product choice especially for children at risk of choline deficiency. The authors propose as mandatory the analysis of all products for their total choline content  based on all contributing components.

Some suggestions:

  1. Line 19 – add please which are the „many secretions”.
  2. Line 21-22: PC is not a lipoprotein. Please reformulate the sentence.
  3. Lines 124-125, you wrote “The plasma levels of choline and its metabolite betaine indicate choline status, which is generally low in the overall population”. What values ​​should these have for infants?
  4. Point 2.1, line 158 – What do you mean by “other nutrient contents?”. Please clarify.
  5. You followed the content of folate, vitamin B12 and long-chain polyunsaturated fatty acids

- taking into account the biodisponibility of Vitamin B12 please specify under which form it is found in these nutritional products

-at discussion please explain in detail why did you choose to follow the content of  folate, vitamin B12 and long-chain polyunsaturated fatty acids

  1. Please remove the numbering from the introduction (1.1...)

Author Response

Reviewer 1:

In the current study the authors investigated the choline content and related nutrients (folate, cobalamin, arachidonic acid and docosahexaenoic acids of frequently used commercial pediatric nutritional products in Germany, namely 105 products. They found that the amount of choline in commercial pediatric nutritional products is heterogenous, particularly for products for children >6 months of age. This may have implications on product choice especially for children at risk of choline deficiency. The authors propose as mandatory the analysis of all products for their total choline content  based on all contributing components.

Some suggestions:

  1. Line 19 – add please which are the „many secretions”.

Thank you for this comment. We have added: “like bile, lipoproteins, surfactant and others” (L.18)

  1. Line 21-22: PC is not a lipoprotein. Please reformulate the sentence.

Thank you! We have changed the sentence to: …(DHA) acid via the phosphatidylcholine moiety of lipoproteins (L.21).

  1. Lines 124-125, you wrote “The plasma levels of choline and its metabolite betaine indicate choline status, which is generally low in the overall population”. What values should these have for infants?

Again, thank you for this comment: We have included reference values in the text (l. 66ff): Reference values of choline are age-dependent, being 41 (32–51)µmol/L in the fetus, 14(10–17)µmol/L in pregnant women, and 9(6–11)µmol/L in non-pregnant women and adults males. Values untimely drop to 22 (16–28)µmol/L in preterm infants, and are only 6 (5–7)mmol/L in CF with exocrine pancreatic insufficiency and TPN patients. Values for betaine are 26(18–39)µmol/L for the fetus and adult, but low in pregnants (11(10-14)µmol/L) and CF patients (19 (15–25)µmol/L) [6,21,27, 29].

  1. Point 2.1, line 158 – What do you mean by “other nutrient contents?”. Please clarify.

Thank you.

We specified to (l. 175f): We collected the information on contents in energy, protein, carbohydrate, choline, folate, cobalamin (vitamin B12), ARA and DHA …

  1. You followed the content of folate, vitamin B12 and long-chain polyunsaturated fatty acids

- taking into account the biodisponibility of Vitamin B12 please specify under which form it is found in these nutritional products

        Thank you for your comment, which is very important. However, unfortunately neither the folate nor the vitamin B12 moiety are generally supplied in the available information. Hence, it is principally not clearly defined whether it is cyano-, hydroxy-, methyl- or – likely - even 5′-deoxyadenosyl-cobalamin. We have added specific information into the Results section (L. 282-286): Notably, although bioavailability differs for different components, such as cyano-, hydroxy- or methylcobalamin, no specific informations were provided here. For folate, it was recorded that Bio Combiotic PRE, HA Combiotic PRE-HA, Bio Combiotic 1+2 and HiPP Comfort (all HiPP GmbH & Co, Pfaffenhofen, Germany) contained 5-methyltetrahydrofolate rather than folate.

-at discussion please explain in detail why did you choose to follow the content of  folate, vitamin B12 and long-chain polyunsaturated fatty acids

        Thank you! We explained that in L.311-315: Moreover, we investigated the other main micronutrients that are related to choline metabolism..

  1. Please remove the numbering from the introduction (1.1...)

Thank you. The numbering was removed from the introduction.

Reviewer 2 Report

Comments and Suggestions for Authors

Interesting idea of ​​this study, my recommendations are the following:
1.6. I recommend that the purpose be mentioned in a single sentence. Supplementary aspects should be mentioned as research objectives or hypotheses.
Methods- I recommend introducing a new subsection called Study design, where the typology of the study and other specific aspects should be mentioned.
I recommend that under table 2, it should be mentioned descriptively what the acronyms represent.
4.3. I recommend that the Perspectives should be clearly mentioned.

Author Response

Reviewer 2:

Interesting idea of this study, my recommendations are the following:
1.6. I recommend that the purpose be mentioned in a single sentence.

Supplementary aspects should be mentioned as research objectives or hypotheses.

Thank you very much for this suggestion. We have changed the last para of the introduction (L.146-150): As choline is a potentially critical nutrient, the aim of this study was to investigate the choline content of frequently used commercial paediatric nutritional products in Germany. Furthermore, as individual nutrient requirements change during development and in response to diseases, this was assessed in relation to total energy and nutrients closely linked to choline metabolism, such as folate, cobalamin, DHA, and ARA.

Methods- I recommend introducing a new subsection called Study design, where the typology of the study and other specific aspects should be mentioned.
Thank you for this suggestion. We have introduced a new para 2.1-Study design (L. 160-172)

I recommend that under table 2, it should be mentioned descriptively what the acronyms represent.

Thank you very much. The meanings of acronyms are explained in a legend (L. 201f)

4.3. I recommend that the Perspectives should be clearly mentioned.

Thank you very much. We have rewritten chapter 5, now clearly pointing to the characterization and quantification of all choline compounds in products (L. 416-426).

Reviewer 3 Report

Comments and Suggestions for Authors

Dear authors, I have reviewed the manuscript titled “Choline in Pediatric Nutrition: Assessing Formula, Fortifiers and Supplements Across Age Groups and Clinical Indications”. Overall, this manuscript is well-structured and presents a valuable evaluation of choline content and related nutrients in pediatric nutritional products, particularly focusing on high-risk groups. However, the following modifications are needed to improve the quality of the manuscript:

Major Comments

  • The abstract should not exceed 250 words. Please try to be more synthetic.
  • In the abstract, consider stating more clearly that biochemical content from non-declared sources (e.g., milk lipids, lecithin) might significantly raise choline content beyond label values.
  • As regards inclusion criteria for products, it's unclear how the 105 products were selected beyond local clinical usage. Was there a systematic sampling or inclusion/exclusion criteria?
  • I would suggest using a more robust statistical testing for comparison (e.g., ANOVA with post-hoc correction across more subgroups).
  • I would suggest adding a graphical diagram (e.g., metabolic pathways) for the discussion of choline metabolism.
  • I would suggest explicitly listing clinical recommendations based on the findings.
  • The reference to the Appendix should be revised (e.g. in line 280 “Appendix Fig. A1A-D” is cited, but it refers to Fig 1 in Appendix B). Additionally, Figure 1 in Appendix B would be more complete if the letters mentioned in the description were added near the graphs.

Minor Comments

  • Please use italics to indicate p-
  • The use of English could be improved. Here some tips:
    • Phrases like "choline may does not match" (in Section 4.2) contain grammatical errors and should be corrected for clarity. Please consider to change it with something like: “The indicated choline may not match the actual content in the products”.
    • Remove the word “to” from the ending sentence (line: 379).
  • From my perspective, there is no need to include the links to the products in the tables. Without them, the table might look cleaner and easier to read.

Author Response

Reviewer 3:

Dear authors, I have reviewed the manuscript titled “Choline in Pediatric Nutrition: Assessing Formula, Fortifiers and Supplements Across Age Groups and Clinical Indications”. Overall, this manuscript is well-structured and presents a valuable evaluation of choline content and related nutrients in pediatric nutritional products, particularly focusing on high-risk groups. However, the following modifications are needed to improve the quality of the manuscript:

        We thankfully acknowledge the reviewer’s appreciation of our manuscript and will follow his/her suggestions.

Major Comments

  • The abstract should not exceed 250 words. Please try to be more synthetic.
    • We have rephrased and shortened the abstract to 247 words.
  • In the abstract, consider stating more clearly that biochemical content from non-declared sources (e.g., milk lipids, lecithin) might significantly raise choline content beyond label values.
    • Thank you very much for this comment. We have rephrased the text and clarified this in l. 31-34.
  • As regards inclusion criteria for products, it's unclear how the 105 products were selected beyond local clinical usage. Was there a systematic sampling or inclusion/exclusion criteria?
    • The selection of products is based on their use according to clinical practice, availability of comparable products for indications, dietitians’ and clinicians’ information within the Pediatric Clinic Tübingen, and the respective patient files as indicated in l. 160-172 (2.1-Study design)
  • I would suggest using a more robust statistical testing for comparison (e.g., ANOVA with post-hoc correction across more subgroups).
    • We had to use non-parametric testing and correction for multiple-group comparison according to Dunn. The text was corrected accordingly (2.4-Statistics, L. 204-20706).
  • I would suggest adding a graphical diagram (e.g., metabolic pathways) for the discussion of choline metabolism.
    • Thank you for this suggestion. We and others have included such diagrams in several publications that are freely available. Here we include a graph, and the respective references for deeper insight (Additional Fig. 4, L. 311-315)
  • I would suggest explicitly listing clinical recommendations based on the findings.
    • Thank you! We have changed the text and included clinical recommendations in the Discussion (L. 382-393).
  • The reference to the Appendix should be revised (e.g. in line 280 “Appendix Fig. A1A-D” is cited, but it refers to Fig 1 in Appendix B). Additionally, Figure 1 in Appendix B would be more complete if the letters mentioned in the description were added near the graphs.
    • These errors have been corrected (L.302; Fig. 1-Appendix B)

Minor Comments

  • Please use italics to indicate p-
    • Thank you! Done
  • The use of English could be improved. Here some tips:
    • Phrases like "choline may does not match" (in Section 4.2) contain grammatical errors and should be corrected for clarity. Please consider to change it with something like: “The indicated choline may not match the actual content in the products”.

Thank you, the paper has been thoroughly corrected for such erreors

    • Remove the word “to” from the ending sentence (line: 379).

Thank you, this was deleted

  • From my perspective, there is no need to include the links to the products in the tables. Without them, the table might look cleaner and easier to read.
    • The links have been removed. Yes, it looks better and anybody can find them easily.

Reviewer 4 Report

Comments and Suggestions for Authors

Can the authors state the principal research question or hypothesis driving this study, and justify why this approach (surveying label-based data) is sufficient to answer it?
Why did the authors not conduct any biochemical analysis of the products to verify the reported choline content, especially when the principal argument is that actual choline content is underreported or unknowable?

Why do the authors support quantification of this information for comparison when they themselves consider that the cited values are not a good indicator of choline content?
Are the authors able to explain why they did not include products manufactured elsewhere and speak to products' geographic limitations' influence on their findings' overallizability?

How do the authors recommend that clinicians interpret these findings in the lack of explicit thresholds for satisfactory choline intake among different pediatric disease states?
On what grounds is the selection of such nutrients in case their interaction with choline metabolism is not thoroughly investigated, and how do their levels affect the study conclusions?

Can the authors better present empirical or clinical evidence to support their claim of lacking adequate current regulatory levels for choline after 6 months beyond reference to their own measurement of plasma levels?
Have the authors considered the practicability, cost, and regulatory problems of implementing such broad-based testing, and do they present any particular policy recommendations or associations to further advance this proposal?

Author Response

Reviewer 4:

We first would like to thank the reviewer for the questions risen, as they enable us to improve our manuscript.

Can the authors state the principal research question or hypothesis driving this study, and justify why this approach (surveying label-based data) is sufficient to answer it?

        Many thanks for this question. The hypothesis is that fortifiers and formula products may increase plasma choline concentrations to different amounts. Hence, first question is to assess and second to optimize choline supply. The most obvious way to assess choline supply in patients is to control for the declared ingredients. As an additional result we found out that this is the case for many products, but not for all of them, requiring systematic analysis or change of legislation. We have clearly written this now in the revised abstract (L.22-23): Fortifiers, formula and supplements may differently impact their choline supply.

Why did the authors not conduct any biochemical analysis of the products to verify the reported choline content, especially when the principal argument is that actual choline content is underreported or unknowable?

        Thank you for this question. As a basis of evaluating choline contents in formula, we assumed that the provided information was correct and complete. It was a finding of this study – while assuming that the amounts of added choline are correct – that information is incomplete.

Why do the authors support quantification of this information for comparison when they themselves consider that the cited values are not a good indicator of choline content?

        This is because information of added choline is clearly an indicator of the minimal amount present and the compound added. Our data may be important for clinicians when choosing a product. For example, MetaX products are free of added choline chloride or bitartrate (which may be advantageous for CF patients who react with strong TMAO increases here. Nevertheless, they contain choline via emulsifiers (phosphatidylcholine), making the product an option. The total amount provided is to be determined in upcoming analyses

Are the authors able to explain why they did not include products manufactured elsewhere and speak to products' geographic limitations' influence on their findings' overallizability?

        This study is a first attempt to characterize and categorize the products regularly used in Tübingen and other German speaking areas and countries. This is, to our opinion, the first step as it concerns ‘our patients’. To extend this to further areas is a matter of future studies, possibly by research units of other areas and markets.

How do the authors recommend that clinicians interpret these findings in the lack of explicit thresholds for satisfactory choline intake among different pediatric disease states?

        We have commented on this in the discussion (L. 382-393), saying that some products without added choline may nevertheless improve choline supply.

On what grounds is the selection of such nutrients in case their interaction with choline metabolism is not thoroughly investigated, and how do their levels affect the study conclusions?

        Thank you. The selection is based on the complete information on locally used or potentially usable products by clinicians, dietitians taking care of patients, nurses and patient files of the Tübingen Children’s Hospital. This is indicated in 2.1 – Study design. L. 160-172

Can the authors better present empirical or clinical evidence to support their claim of lacking adequate current regulatory levels for choline after 6 months beyond reference to their own measurement of plasma levels?

        Unfortunately, there is no clinical study here. These is only the IoM data on age-specific requirements, and the lack of added choline in several products according to legislation.

Have the authors considered the practicability, cost, and regulatory problems of implementing such broad-based testing, and do they present any particular policy recommendations or associations to further advance this proposal?

        Compared to the numbers of packages or bottles in a single batch, the costs of determining choline components (at our laboratory the costs would be about 250€ including overhead and VAT) seems negligible (L.402).

Reviewer 5 Report

Comments and Suggestions for Authors

Major:

  1. Authors should clearly state earlier in the manuscript that no direct biochemical choline measurements were performed, and explicitly identify this as a study limitation.
  2. Authors should elaborate on the clinical implications of their findings by providing more specific guidance or recommendations for pediatric care, especially in at-risk populations such as preterm infants and children with cystic fibrosis.
  3. Authors should enhance the discussion by comparing regulatory approaches to choline supplementation internationally to better contextualise the current guidelines' limitations.

Minor:

  1. Authors should consider refining the title for clarity, e.g., "Assessment of Choline Content in Pediatric Nutritional Products." The abstract could also be slightly more concise and directive.
  2. Authors should revise several sentences for improved readability and clarity. In particular, lengthy or complex constructions should be simplified.
Comments on the Quality of English Language

Minor linguistic and stylistic corrections are required.

Author Response

  1. Authors should clearly state earlier in the manuscript that no direct biochemical choline measurements were performed, and explicitly identify this as a study limitation.

Thank you! We have clearly stated this in the text (L. 25)

  1. Authors should elaborate on the clinical implications of their findings by providing more specific guidance or recommendations for pediatric care, especially in at-risk populations such as preterm infants and children with cystic fibrosis.

Thanks again. We have commented on this in the discussion (L. 382-393), saying that some products without added choline may nevertheless improve choline supply.

  1. Authors should enhance the discussion by comparing regulatory approaches to choline supplementation internationally to better contextualise the current guidelines' limitations.

Thank you, again. We have done that in L. 140-143.

Minor:

  1. Authors should consider refining the title for clarity, e.g., "Assessment of Choline Content in Pediatric Nutritional Products." The abstract could also be slightly more concise and directive.

Thank you! We have rewritten the abstract, now containing 250 words

  1. Authors should revise several sentences for improved readability and clarity. In particular, lengthy or complex constructions should be simplified.

Thank you very much! The text has been completely revised, most parts have been rewritten and were checked by authors for orthography, grammar and style.

Round 2

Reviewer 3 Report

Comments and Suggestions for Authors

I have reviewed the revisions made by the authors in response to the comments provided. The authors have made the necessary changes to improve the quality of the work that know is suitable for acceptance in its current form.

Reviewer 4 Report

Comments and Suggestions for Authors

The paper can be accepted in its present form.